# Abnormal center of mass feedback responses during balance: A potential biomarker of falls in Parkinson's disease

J. Lucas McKay[1,2,3]*, Kimberly C. Lang[4], Sistania M. Bong[3], Madeleine E. Hackney[5,6], Stewart A. Factor[2], Lena H. Ting[1,7]

1 Department of Biomedical Informatics, Emory University School of Medicine, Atlanta, Georgia, United States of America, 2 Jean & Paul Amos PD & Movement Disorders Program, Department of Neurology, Emory University School of Medicine, Atlanta, Georgia, United States of America, 3 Wallace H. Coulter Department of Biomedical Engineering, Emory University and Georgia Tech, Atlanta, Georgia, United States of America, 4 Graduate Division of Biological and Biomedical Sciences, Emory University, Atlanta, Georgia, United States of America, 5 Department of Medicine, Division of General Medicine and Geriatrics, Emory University School of Medicine, Atlanta, Georgia, United States of America, 6 Rehabilitation R&D Center, Atlanta VA Medical Center, Decatur, Georgia, United States of America, 7 Department of Rehabilitation Medicine, Division of Physical Therapy, Emory University, Atlanta, Georgia, United States of America

* lucas@dbmi.emory.edu

**Data Availability Statement:** All relevant data are within the paper and its Supporting information files. Participant-level and timecourse data are

## Abstract

Although Parkinson disease (PD) causes profound balance impairments, we know very little about how PD impacts the sensorimotor networks we rely on for automatically maintaining balance control. In young healthy people and animals, muscles are activated in a precise temporal and spatial organization when the center of body mass (CoM) is unexpectedly moved that is largely automatic and determined by feedback of CoM motion. Here, we show that PD alters the sensitivity of the sensorimotor feedback transformation. Importantly, sensorimotor feedback transformations for balance in PD remain temporally precise, but become spatially diffuse by recruiting additional muscle activity in antagonist muscles during balance responses. The abnormal antagonist muscle activity remains precisely time-locked to sensorimotor feedback signals encoding undesirable motion of the body in space. Further, among people with PD, the sensitivity of abnormal antagonist muscle activity to CoM motion varies directly with the number of recent falls. Our work shows that in people with PD, sensorimotor feedback transformations for balance are intact but disinhibited in antagonist muscles, likely contributing to balance deficits and falls.

## Introduction

Parkinson's disease (PD) causes profound balance impairments and falls, but we still know surprisingly little about how PD affects the ways in which motor outputs during balance control are organized in the nervous system based on incoming sensory information. When people maintain upright standing balance, incoming sensory information about body motion is processed by the nervous system to generate motor commands sent to activate muscles throughout the body [1]. We use the term "sensorimotor transformation" to describe this ongoing

available in an online repository: https://doi.org/10.7910/DVN/2XIUFP.

**Funding:** This study was supported in part by National Institutes of Health grants K25HD086276, R01HD046922, R21HD075612, UL1TR002378, UL1TR000454; Department of Veterans Affairs R&D Service Career Development Awards E7108M and N0870W, The Consolidated Anti-Aging Foundation, the APDA center for Advanced Research at Emory University, and the Sartain Lanier Family Foundation. The funders had no role in study design, data collection and analysis, decision to publish, or preparation of the manuscript.

**Competing interests:** The authors have declared that no competing interests exist.

process, where sensory signals arising during a loss of balance are interpreted and used to shape the resulting balance correcting motor signals [2]. Here, our goal was to investigate whether and how sensorimotor transformations for reactive balance responses, i.e. the sensitivity of evoked muscle activity to sensory signals encoding balance error, are altered in PD.

One clue that sensorimotor balance transformations might be affected in PD is that–in addition to the fact that PD significantly increases fall risk [3]–falls in people with PD tend to occur in specific conditions involving the control of the center of body mass (CoM) that are distinct from those of falls in the general aging population. Maintaining the CoM over the base of support is critical for balance control, and dysregulation of either the CoM or the base of support can cause a fall [4, 5]. However, while older adults fall most frequently due to slips and trips that cause a sudden change in the area of the base of support, people with PD fall most frequently during activities that require them to actively control the CoM [3] and during which there is no change on the base of support. Examples include turning around while indoors [6] or backward retropulsion while rising from a chair [7]. In a recent study of actual fall events captured in video recordings of individuals in long-term assistive care, people with PD were significantly more likely to experience falls provoked by incorrect weight shifting [8]. Such activities require that the nervous system sense and monitor the body and generate appropriate motor responses, suggesting that there might be impairments in sensorimotor balance transformations that are more pronounced in PD than in the general geriatric community.

Laboratory studies have also shown that motor signals to muscles are abnormal in people with PD. When muscles are stretched and lengthened, spinal reflexes typically increase the activity in the muscle, helping to maintain limb posture. (These muscles are often referred to as "agonists.") Work in the early- to mid-1900s described paradoxical "shortening reactions" in PD, in which muscles activate when they are shortened during passive movements, counteracting the stretch reflex response and reinforcing the new imposed position [9]. Later studies showed that PD patients exhibit a complex pattern of motor signals during voluntary reaching, in which agonist and antagonist muscles activate in series of multiple bursts rather than the simple three-burst pattern seen in people without PD [10]. Studies using moving platforms to perturb standing balance show the main responses for balance corrections during support-surface translations are in agonist muscles that are first stretched by the balance perturbations. However, in PD, the antagonist muscles that are shortened by perturbations in PD patients are also activated, counteracting the corrective torques generated by the agonist muscles lengthened by balance perturbations [11–15].

In animals and healthy young human subjects, motor signals to agonist or "prime mover" muscles are generated based on sensory feedback of CoM motion during balance. However, it is unclear whether this sensorimotor transformation is preserved in PD. When standing balance is disturbed, motor signals to agonist muscles are created with a sensorimotor feedback transformation in which agonist muscles are activated in proportion to CoM motion [2, 16–18]. Patterns of magnitude and timing of muscle activation can be explained with a small set of feedback gains that describe sensitivity to CoM acceleration, velocity and displacement ($k_a$, $k_v$, and $k_d$); and a delay ($\lambda$) to account for sensorimotor processing and transmission time [16–18]. We know from animal studies that this activity depends on brainstem and spinal networks, with important roles for subcortical structures including thalamus and subthalamic nucleus (STN) [19–22]. What remains unclear is whether the sensorimotor transformation for balance control is disrupted in PD, but clearly these subcortical structures, i.e. the thalamus and STN, are affected [23]. Further, even if CoM feedback is abnormal in PD the relevance to functional outcomes like falls is unknown. For example, PD is associated with additional coactivation during walking, but the presence of coactivation is weakly related to gait quality [24].

The sensorimotor processing underlying motor signals to antagonist muscles during balance are not understood, but may arise from similar CoM sensorimotor feedback signals driving agonist muscles. An antagonist activation pathway based on CoM feedback analogous to that of agonist muscles seems plausible, as antagonist muscles are frequently activated during motor tasks in uncertain environments [25–27]. If so, paradoxical antagonist activity in PD may be generated by otherwise healthy sensorimotor processes, for example, processes that have been released from tonic inhibition [28] or activated by descending signals from higher motor centers with abnormal timing, [29] given that basal ganglia dysfunction can produce both hypo–and hyper–kinetic signs [30]. In one previous study, a CoM feedback scheme was used to explain changes in components of antagonist muscle responses over the course of motor adaptation; [17] however, whether antagonist activity during balance in general or in the parkinsonian state can be described by CoM feedback is unknown.

Here, we tested how sensorimotor transformations both driving agonist and antagonist muscles during balance control are disrupted in PD, and related these changes to falls as well as clinical variables reflecting different aspects of the underlying pathophysiology. Using support-surface perturbations to standing balance, we disturbed the position of the CoM over the base of support. We tested whether sensorimotor feedback transformation identified previously in younger adults could explain the generation of both agonist and antagonist muscle activity in people with PD and age-similar older adults without PD (Non-PD), and whether there were abnormalities specific to PD. We further tested whether the presence and severity of abnormal CoM feedback was associated with the presence and number of previous falls over the prior 6 months. Finally, we tested whether the abnormal sensorimotor transformation features were associated with clinical variables reflecting PD severity and duration, and with clinical measures of balance impairment.

Overall, we found that sensorimotor transformation from CoM sensory information to motor signals to *tibialis anterior* was elevated in PD vs. Non-PD, with increased amounts of feedback and a longer processing time (by ≈20 ms) than the sensorimotor transformation from CoM to agonist muscles. Further, this abnormal sensitivity to CoM acceleration was associated with a history of frequent falls. Taken together, abnormal CoM feedback likely contributes to balance impairments and may arise from abnormal activity of supraspinal centers that interact with brainstem and spinal networks previously hypothesized to mediate sensorimotor transformations for balance in young healthy individuals and animals.

## Results

### Participants and setting

We examined temporal patterns of muscle activation evoked by support surface translation perturbations in N = 44 people with mild-moderate Parkinson disease (PD, average disease duration, 7.4 ± 4.8 y, range 8 m-21 y) and N = 18 matched neurotypical individuals (Non-PD, Table 1). Most PD patients (62%) were of the Postural Instability and Gait Disability (PIGD) PD symptom phenotype, [31] and most (66%) did not report freezing of gait (FOG). With the exception of 2 PD patients early in the disease course who had not yet started pharmacotherapy, the remainder were prescribed antiparkinsonian medications (average levodopa equivalent daily dose (LED) [32] 726 ± 357 mg, range 280–1800 mg). All assessments were performed in the practically-defined 12-hour OFF medication state [11, 33]. None had previously undergone functional neurosurgery. Detailed medication information is provided in S1 File.

No statistically-significant differences were observed between PD and Non-PD participants on age, sex, height, weight, or overall cognition (Montreal Cognitive Assessment, MoCA [34]). Participants were cognitively normal according to established criteria [35]. However, PD

**Table 1. Demographic and clinical characteristics of PD and age-matched Non-PD participants.**

| Variable | PD | Non-PD | Total |
|---|---|---|---|
| N | 44 | 18 | 62 |
| Age, y | 68 ± 7 | 66 ± 8 | 68 ± 7 |
| Sex | | | |
| Male, n (%) | 23 (52) | 7 (39) | 30 (48) |
| Female, n (%) | 21 (48) | 11 (61) | 32 (52) |
| Height, cm | 169 ± 10 | 167 ± 11 | 169 ± 10 |
| Weight, kg | 76 ± 14 | 76 ± 15 | 77 ± 14 |
| MoCA, /30 | 27.8 ± 1.5 | 27.2 ± 1.4 | 27.6 ± 1.5 |
| PD duration, y | 7.4 ± 4.8 | | |
| MDS UPDRS-III, /132 | 36.1 ± 13.0 | | |
| LED, mg[a] | 726 ± 357 | | |
| Freezing of Gait | | | |
| Nonfreezer, n (%) | 29 (66) | | |
| Freezer, n (%) | 15 (34) | | |
| PD Phenotype | | | |
| PIGD, n (%) | 27 (61) | | |
| TD, n (%) | 13 (30) | | |
| Indet, n (%) | 4 (9) | | |
| Balance outcomes | | | |
| MiniBESTest, /28[b,†] | 22.1 ± 4.1 | 24.7 ± 1.4 | 22.8 ± 3.7 |
| FAB, /40[c,†] | 29.8 ± 5.5 | 33.2 ± 2.7 | 30.9 ± 5.0 |
| DGI, /24[d,†] | 20.4 ± 3.5 | 22.3 ± 1.3 | 21.0 ± 3.1 |
| BBS, /56[d,†] | 52.9 ± 5.5 | 55.0 ± 1.3 | 53.6 ± 3.4 |

**Abbreviations**: PD, Parkinson disease; MoCA, Montreal Cognitive Assessment; UPDRS-III, Unified Parkinson Disease Rating Scale, Part III: Motor Exam; LED, levodopa equivalent daily dose; PIGD, Postural Instability/Gait Difficulty; TD, Tremor Dominant; Indet., Indeterminate; BBS, Berg Balance Scale; FAB, Fullerton Advanced Balance Scale; DGI, Dynamic Gait Index.

[a]N = 41. Two patients had not yet begun pharmacotherapy and dosage information was unavailable for one patient who was prescribed carbidopa/levodopa. LED was calculated as in previous studies [32] with conversion factor of 0.6 assumed for Rytary.

[b]N = 28.

[c]N = 31.

[d]N = 30. Demographic information for the reference sample of young healthy participants (N = 6) is provided in the main text.

participants exhibited poorer overall performance on clinical balance and gait outcomes, with statistically-significant impairment on the current clinical standard MiniBESTest [36–38] (2.6 points, P = 0.017, independent samples *t*-test, unpooled variance assumption) as well the Fullerton Advanced Balance Scale (FAB, [36] 3.4 points, P = 0.028), the Dynamic Gait Index (DGI, [39] 1.9 points, P = 0.043), and the Berg Balance Scale (BBS, [40] 2.1 points, p = 0.040). Some analyses also considered a reference sample of N = 6 young healthy individuals recruited from a college campus (HYA, 4 female, 21 ± 2 y). No statistically-significant differences were identified between groups on sex, height, MoCA score; however, younger participants weighed less (61 ± 12 vs. 77 ± 14 kg; P = 0.039).

## In PD, balance-correcting muscle activity is normal, but antagonist TA muscle activity is elevated

We assessed each participant with support surface translation perturbations delivered in unpredictable order in directions evenly spaced throughout the horizontal plane (Fig 1A). The

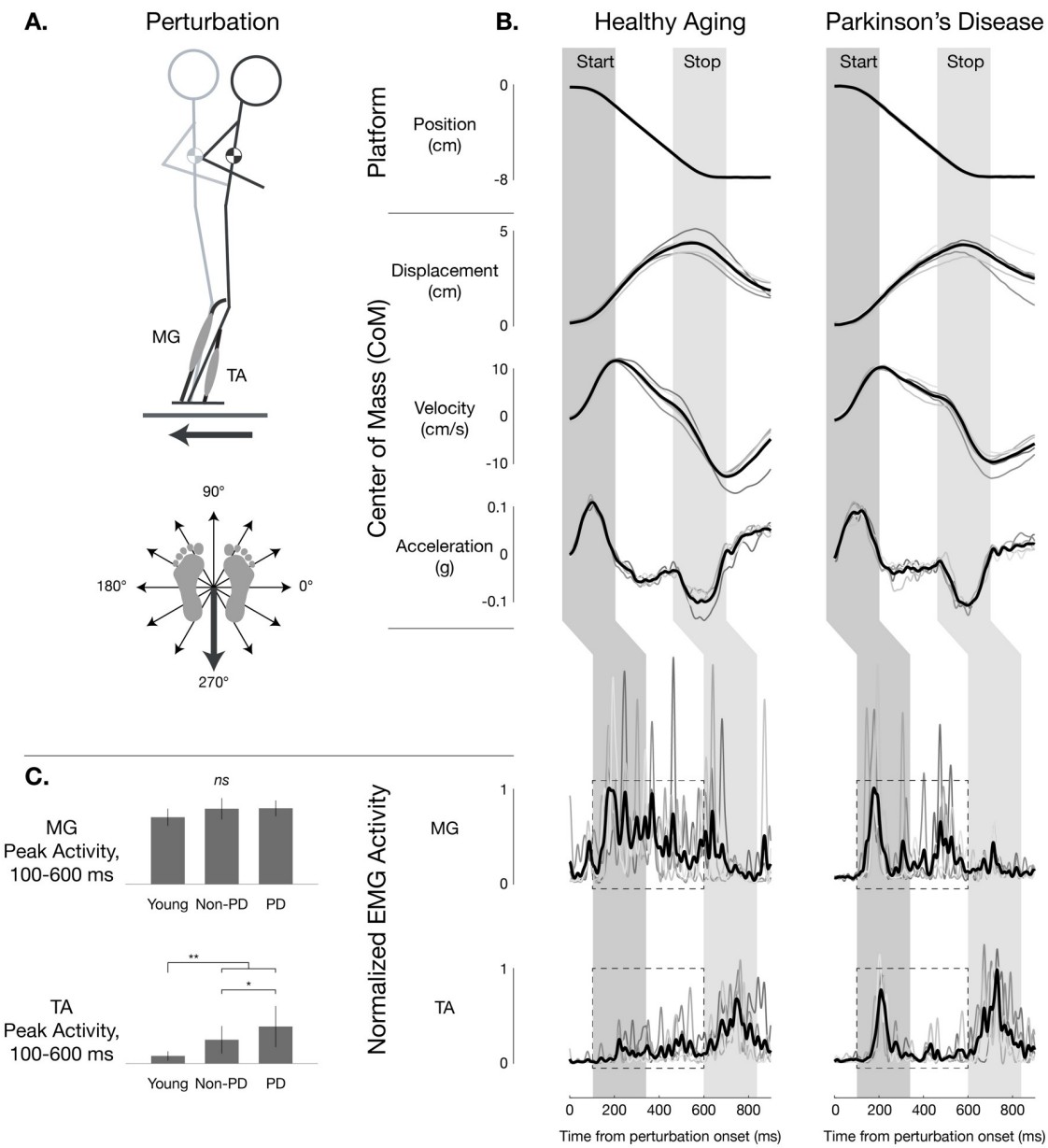

**Fig 1. Postural responses in healthy aging and in Parkinson disease.** A: Schematic depiction of testing paradigm. B: Representative postural responses in healthy aging and Parkinson disease. Left: a healthy older woman (age, 69 y; no recent falls; MiniBESTest score 23/28). Right: an older woman with PD (age, 73 y; disease duration, 20 y; 1 fall in the previous 6 months; MiniBESTest score 24/28; MDS-UPDRS-III OFF score 26/132; complaints of freezing of gait (FoG); LEDD 675 mg). In response to a backward (270˚) perturbation, the center of mass (CoM) initially moves in the forward direction (*start* epoch) such that the CoM kinematic signals [displacement (*d*), velocity (*v*), acceleration (*a*)] are all opposite the direction of platform motion. When the platform decelerated (*stop* epoch), CoM acceleration quickly reversed direction to be opposite that of CoM velocity and displacement. Activity of both MG and TA muscle initiated about 100ms after the onset of perturbation in and varied in direct response to the preceding CoM kinematics signals. In both healthy and PD participants, the medial gastrocnemius (MG) activated in the *start* epoch and continued throughout the perturbation; MG is considered the agonist for restoring balance. However, in the healthy individual, there was little activation of tibialis anterior (TA) muscle in the healthy participant, but a marked burst in the TA in the *start* epoch, where TA is considered an antagonist for restoring balance. In both the healthy and PD participant, TA was activated in the *stop* epoch, where the TA is an agonist for balance correction and MG is the antagonist. Overlaid traces shown are for individual trials (5 total). Averages of all traces are shown in black.

apparatus and testing paradigm have been described previously [11]. In order to assess the balance control system without inducing actual falls, the perturbation parameters were refined over a series of studies in Parkinson's patients in both the OFF and ON medication states to be very near, but to not exceed, the level at which they could maintain balance without stepping [11, 41–43]. We analyzed backward platform translation trials (240˚, 270˚, 300˚) that initially lengthened the plantarflexor *medial gastrocnemius* (MG) (Fig 1B, *start* epoch). This is considered a balance-correcting response; MG activity was normalized to the maximal response observed in these trials. We often observed a coactivation response in the dorsiflexor *tibialis anterior* (TA), which is considered to be an antagonist until the end of the perturbation when the platform decelerates (Fig 1B, *stop* epoch). TA muscle activity was normalized to the maximal activity observed in forward perturbations (60˚, 90˚, 120˚) where it acts as the agonist for the primary balance-correcting response [11].

Balance-correcting MG responses to backward translation perturbations in older adults with and without PD were very similar to that observed previously in young healthy individuals [11, 16–18] and in cats, [2] activating when the MG muscle is stretched. MG exhibited an initial burst at latency ≈100–150 ms following the onset of platform motion and initial acceleration of the CoM followed by a plateau of activity of ≈200–400 ms duration comparable to the duration of time in which the CoM was being displaced with positive velocity forward with respect to the ankles (Fig 1B, left).

Individuals with PD also often exhibited a large initial burst of antagonist TA muscle activity during backward perturbations that oppose the balance-correcting actions of the MG muscle (≈100–150 ms) when it was shortened by perturbations (Fig 1B, right). In neurotypical older participants, the TA response to posterior perturbations was typically quite similar to that previously observed in young healthy individuals, and was primarily characterized by a response to the *deceleration* of the platform ≈600 ms after perturbation onset, with little if any activity above background levels during earlier phases. However, in PD patients, the TA response was often characterized by a strong initial burst at ≈100–150 ms latency very comparable to that observed in MG, so that in some cases TA appeared to activate in a pattern of magnitude and timing almost identical to that of MG.

Abnormal antagonist TA activity during backward perturbations was significantly elevated in PD compared to matched neurotypical individuals (56%, P<0.05, ANOVA, post hoc tests; Fig 1C) as well as to young healthy individuals (P<0.01). Difference in peak muscle activity 100–600 ms after perturbation onset were only found in antagonist TA activity, and not balance-correcting MG activity.

Although we have previously reported excessive MG antagonist activity during balance tasks using a similar paradigm in PD patients in the ON medication state, [41] we did not find that MG antagonist activity was elevated consistently across patients in this sample. Therefore, we did not attempt to analyze the timecourse of MG antagonist activity. Possible explanations for this finding and examples of patients with and without elevated MG antagonist activity are presented in S1 File.

## Hypothesized sensorimotor feedback pathways decomposed balance correcting and antagonist muscle activity in healthy aging and in PD

We hypothesized that balance correcting and antagonist muscle activity across aging and disease could be explained by a common underlying sensorimotor transformation between CoM motion and muscle activation. To explicitly test our hypothesis, we reconstructed the entire timecourse of balance correcting and antagonist muscle activity using a model that was

previously used to reproduce balance-correcting muscle activity in healthy and impaired animals as well and in healthy young humans [2, 16–18] (Fig 2).

In the sensorimotor response model (SRM), balance-correcting muscle activity (Fig 2A and 2B, green) is reconstructed by a weighted sum of horizontal plane CoM acceleration ($a$), velocity ($v$), and displacement ($d$) occurring ≈100 ms earlier, which acts to stretch the muscles. Thus, three feedback gain parameters (or weights, $k_a$, $k_v$, $k_d$) and a lumped time delay ($\lambda$) for each muscle are identified by minimizing the error between recorded and reconstructed EMG signals. The resulting parameters quantify the contributions of acceleration, velocity, and displacement sensory signals to balance correcting responses.

It was evident that the established SRM would be unable to explain antagonist TA activity that occurs while it is shortening, rather than stretching (cf. [17]). To explicitly account for abnormal antagonist TA activity in PD during backward perturbations, we extended the SRM with a new antagonist pathway in which sensory signals driving balance-correcting MG activity also activate TA (Fig 2A and 2B, red). TA activation during shortening has been previously reported in our laboratory when younger adults were exposed to novel or unpredictable perturbations [17]. Here, we extended the SRM model to also include parameters associated with the activation of TA as an antagonist ($k_a'$, $k_v'$, $k_d'$, $\lambda'$). These parameters explicitly dissociate hypothesized sensory signals underlying the initial, antagonist TA muscle activity from later, balance-correcting TA muscle activity when the support-surface decelerates at the end of the perturbation.

The addition of the antagonist pathway significantly improved the ability of the SRM to explain PD antagonist TA activity, improving adjusted coefficient of determination ($R_a^2$) by 0.18±0.15 on average (P<0.001) and increasing peak antagonist activity 100–600 ms after perturbation onset by 0.23±0.11 normalized units (nu) on average (P<0.001). When we performed similar comparisons on PD MG antagonist activity during forward perturbations, we noted similar improvements in fits (improvement in $R_a^2$, 0.13±0.14, P<0.001; increase in early antagonist activity, 0.14±0.10 nu, P<0.001); however, antagonist activity remained significantly higher in TA (64%, P<0.001) than in MG (S1 File).

The extended SRM accounted for both balance-correcting and antagonist muscle activity in during backward perturbations with long-latency delayed feedback of CoM kinematics, with grand mean VAF of 82±6% and 81±7% in TA and MG, respectively. These values were generally comparable (attenuated by ≈6% and ≈1%, respectively) to those in an earlier study of young healthy participants [16] and were considered acceptable. The only statistically-significant differences in VAF identified across groups was a small decrease (1%) among young compared to older participants in TA (S1 File).

## TA antagonist CoM acceleration feedback is increased in PD and in aging

We hypothesized that the large initial burst of antagonist TA muscle activity during backward perturbations reflected increased sensitivity to peak CoM acceleration. We observed no meaningful differences in peak CoM acceleration (-3%, P = 0.18, ANOVA), velocity (+7%, P = 0.07), or displacement (-3%, P = 0.73) across groups (S1 File). Therefore, the initial burst could only be reconstructed by increasing the sensitivity of the muscle activity to the initial acceleration.

Consistent with our hypothesis, ANOVA and post-hoc tests showed that TA antagonist CoM acceleration feedback parameter $k_a'$ was significantly higher among PD compared to Non-PD (increased by 95%, P<0.001) as well as among older participants (PD and Non-PD) compared to the young group (+239%, P<0.001) (Fig 2C). Among the PD group, there was no

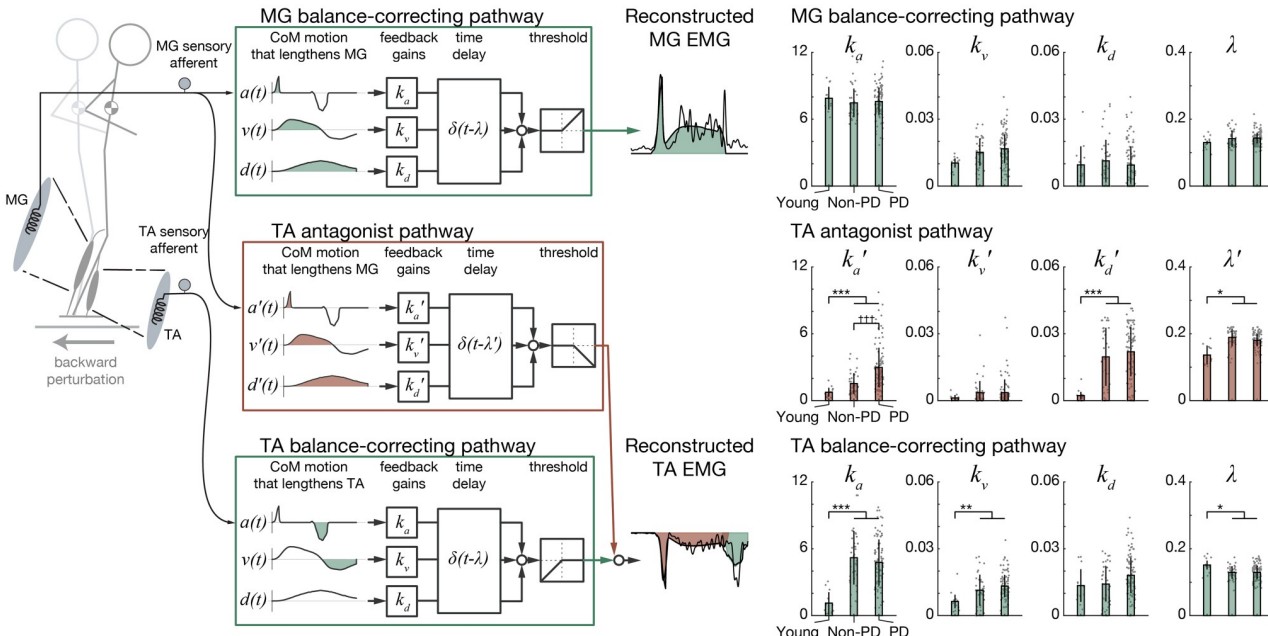

**Fig 2. Reconstruction of balance EMG with combinations of balance-correcting and nonspecific CoM feedback.** A: Representative CoM feedback responses to backward support surface perturbations over healthy aging and in Parkinson disease. Left: a healthy young woman recruited from a college campus (age, 22 y). Middle: a healthy older man (age, 64 y; no recent fall history). Right: an older woman with PD (age, 62 y, disease duration, 5 y; 1 fall in the previous 6 months; MDS-UPDRS-III OFF score 55/132; no complaints of FoG; LEDD 900 mg). Above: MG; Below: TA (reversed scale). Thin lines represent are average EMG (typically 5 trials). Thick lines represent SRM fits. Green and red shaded areas represent balance-correcting and nonspecific SRM feedback. Inset: variance accounted for (VAF) and $R^2$ values indicating goodness of fit. B: Feedback models for balance control. In the model formulation, MG responded to balance-correcting forward-directed kinematic signals only (green) and TA responded to balance-correcting backward-directed (green) and nonspecific forward-directed (red) kinematic feedback. C: Comparison of model parameters across groups. Individual participant data are shown as separate gray dots for each of the left and right legs. ***P<0.001, **P<0.01, *P<0.05, older vs. younger, ANOVA, post-hoc tests. †††P<0.001, PD vs. Non-PD, ANOVA, post-hoc tests.

statistically-significant variation in $k_a'$ across PD phenotypes (P = 0.94, ANOVA on PIGD vs. TD vs. Indeterminate; See S1 File).

Taken together with imaging studies in older adults with and without PD, the elevated values of $k_a'$ in PD suggest that–in addition to the dopaminergic degeneration characteristic of PD–patients with high values of $k_a'$ likely also had substantial cholinergic deficits.

### Other TA CoM feedback parameters are affected in aging

Several other SRM gain parameters varied strongly across age groups, with significantly higher values of antagonist displacement gain $k_d'$ (+854%, P<0.001), as well as of balance correcting acceleration gain $k_a$ (+334%, P<0.001) and of balance correcting velocity gain $k_v$ (+100%, P = 0.003) among the older participants compared to the young group. Compared to the young group, among the older participants values of antagonist delay $\lambda'$ were significantly longer (183 vs. 136 ms, P = 0.018) but values of balance correcting delay $\lambda$ were significantly shorter (129 vs. 151 ms, 14% P = 0.014). No statistically-significant differences among groups were observed in SRM parameters describing balance correcting activity in TA during perturbations that caused it to lengthen, or in MG activity. (See S1 File).

### TA antagonist CoM acceleration feedback is increased with fall history and with number of previous falls

Multivariate analyses showed that SRM parameter $k_a'$ was strongly associated with the presence and number of previous falls. Previous falls were significantly more prevalent among the PD group than among other participants (47% vs. 12%; P = 0.008; compare blue to brown bars, Fig 3A). Compared to participants with no fall history, $k_a'$ was significantly increased (115%, P<0.001, ANOVA; Fig 3B1) among those with ≥2 falls in the prior 6 months after controlling for age, sex, presence of PD, and presence of FOG. $k_a'$ was also significantly increased among participants with PD (37%, P = 0.042; Fig 3B2) and tended to increase with age (≈8%/decade, P = 0.116) in multivariate analysis. No significant effects of sex (P = 0.742) or presence of FOG (P = 0.584) were identified. No significant differences were identified in $k_a'$ between participants with 1 and 0 fall over the prior 6 months (P = 0.721).

Although a maximum of only 1 fall over the prior 6 months was reported in the Non-PD group, average fall frequency was substantially higher in the PD group. The average 6 month fall frequency was 12, and the maximum was 180, corresponding roughly to biweekly and

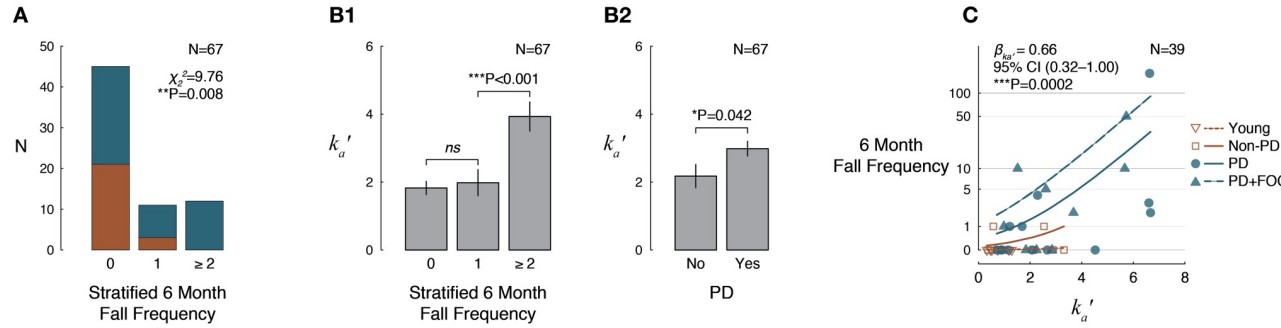

**Fig 3. Associations between TA antagonist acceleration feedback parameter $k_a'$ and fall history over the prior 6 months.** A: Histogram of stratified fall frequency in participants with PD (blue) and without PD (brown). P value reflects $\chi^2$ test of homogeneity. B1,2: Estimated marginal means (±SEM) describing effects of (B1) fall history classification and (B2) PD on $k_a'$, controlling for age, sex, and presence of FOG. P values reflect Wald $\chi^2$ tests. C: Negative binomial regression results describing association between frequency of falls over the 6 months prior to study enrollment and $k_a'$, controlling for age, sex, presence of PD, and presence of FOG. Trendlines indicate expected fall frequencies as a function of $k_a'$ for different simulated cases: Young female (20 y), brown dotted. Older female (68 y), brown, solid. Older female (68 y) with PD but without FOG, blue, solid. Older female (68 y) with PD and FOG, blue, dashed. P value reflects a Wald $\chi^2$ test.

daily falls. The long tail of the fall frequency distribution provided strong support for the use of a negative binomial model to represent the data [44]. Negative binomial regression is common in epidemiological studies and has been frequently used for fall frequency data [45–47]. Multivariate negative binomial regression identified a strong nonlinear relationship between fall frequency and $k_a'$, even after controlling for known fall risk factors (Fig 3C). The identified regression coefficient between fall frequency and $k_a'$ was $\beta_{ka'}$ = 0.66 (95% CI 0.32–1.00; P = 0.0002). The model indicated that although among young participants an increase in $k_a'$ from the 15th to 85th percentile had no effect on the expected number of falls (0), among the older Non-PD group, a similar increase in $k_a'$ was associated with an increase in the expected number of falls from 0 to 3, and among PD, a similar increase in $k_a'$ was associated with an increase in the expected number of falls from 1 to 11. Estimated fall frequencies as functions of $k_a'$ for different estimated participants are shown in Fig 3C.

## TA antagonist CoM acceleration feedback is most strongly associated with disease severity and reactive balance on MiniBESTest

After controlling for effects of age, $k_a'$ was most strongly associated with clinical indicators of disease severity and with a behavioral measure of reactive postural control on the MiniBESTest (Fig 4). Given the sample size and limited emphasis on reactive postural control in the MDS-UPDRS-III–and to a lesser extent, MiniBESTest, both of which are considered inadequate for assessing fall risk [37]–we did not expect to identify strong correlations with clinical variables. In particular, ≈40% of MDS-UPDRS-III items measure tremor severity or upper limb motor performance, which are not commonly considered to be associated with falls.

We considered Pearson product-moment correlation coefficients ≥0.10 to be non-negligible, according to criteria proposed by Cohen [49]. Univariate models (Fig 4A) identified non-negligible correlations between $k_a'$ and clinical measures of disease severity and reactive balance, including: increased age ($r$ = 0.18); increased disease severity as indicated by increased disease duration ($r$ = 0.26), amount of dopaminergic medication (LED; $r$ = 0.19), total symptoms (MDS-UPDRS-III total score; $r$ = 0.19); and more impaired postural control on MDS-UPDRS-III ($r$ = 0.10) and MiniBESTest ($r$ = -0.16).

It was notable that no association was identified between $k_a'$ and overall balance ability as indicated by total MiniBESTest score ($r$ = -0.02), potentially because most patients were quite high-performing on this test, with >75% above clinical cutoff values for fall risk [35]. More severe $k_a'$ was also associated with more impaired cognition on the MoCA ($r$ = -0.12), which was notable given that >95% of the sample MoCA scores were ≥26, indicative of normal cognition in PD [50]. After adjusting for effects of age, the only associations that remained non-negligible were with PD duration, LED, and MiniBESTest reactive postural control (Fig 4B, crosses). No identified correlations were statistically significant.

## TA antagonist CoM acceleration feedback delay times are consistent with long-loop activity

The delay between CoM kinematics and muscle activity in the destabilizing pathway was substantially longer than in the stabilizing pathways, suggesting the involvement of higher-level neural influences (Fig 5). Identified SRM delay parameters in each of the identified pathways were not different in PD versus non-PD groups (p = 0.54). The TA-TA stabilizing pathways had delay parameters that were about 15 ms shorter (129±20 ms) than the MG-MG (143±19) stabilizing pathway (P<0.001). The MG-TA destabilizing pathway was >40 ms longer than both stabilizing pathways (183±21, P<0.001). This additional 40+ ms loop time for the MG-TA destabilizing pathway is sufficiently long for basal ganglia involvement, [51]

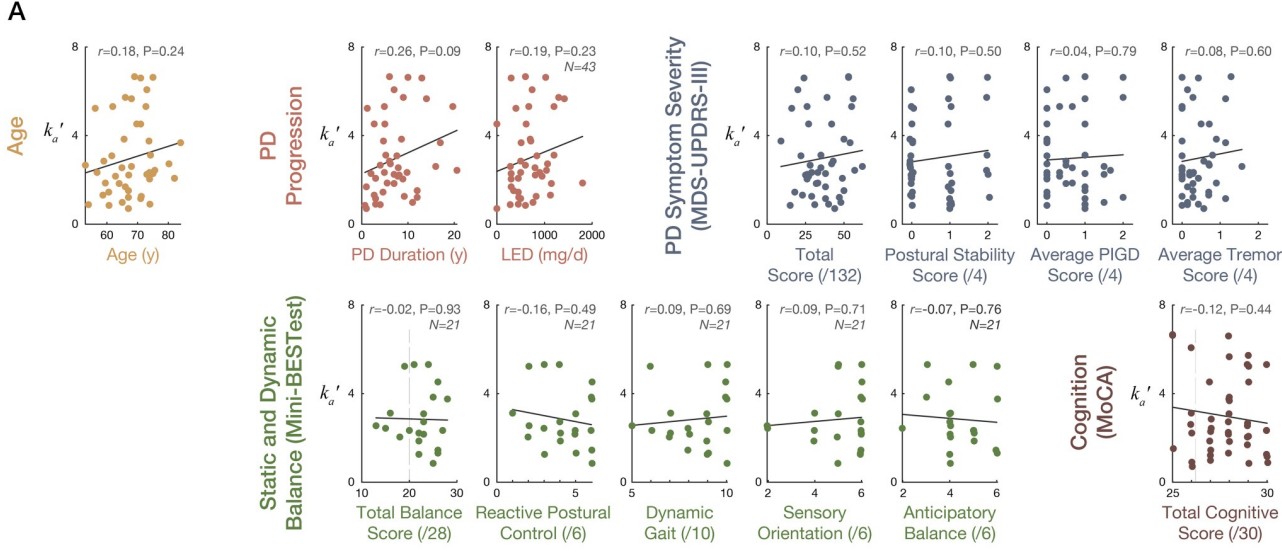

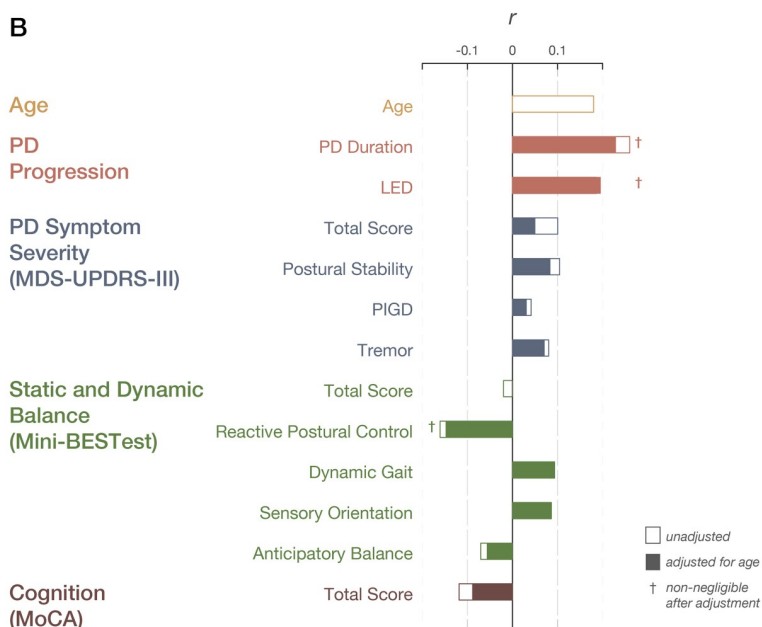

**Fig 4. Associations between antagonist acceleration feedback parameter $k_a'$ and clinical and demographic variables among participants with PD.**
A: Associations between SRM parameters and clinical and demographic variables. Solid lines indicate best-fit linear regressions. N = 44 except as noted.
Vertical lines indicate: MiniBESTest scores <20, proposed cutoff value for 6-month fall risk in people with PD; [48] average MoCA scores for
cognitively-normal people with PD (26±3) [35]. B: Pearson product-moment correlation coefficients calculated between $k_a'$ and clinical variables. Open
bars indicate univariate correlation coefficients. Closed bars indicate correlations corrected for age effects.

compared to the shorter loop time of the MG-MG stabilizing pathway that is hypothesized to
arise from reticulospinal circuits [2].

## Discussion

Here we demonstrate that in PD, sensorimotor transformations during perturbations to stand-
ing balance are intact but dysregulated, generating temporally precise but spatially diffuse

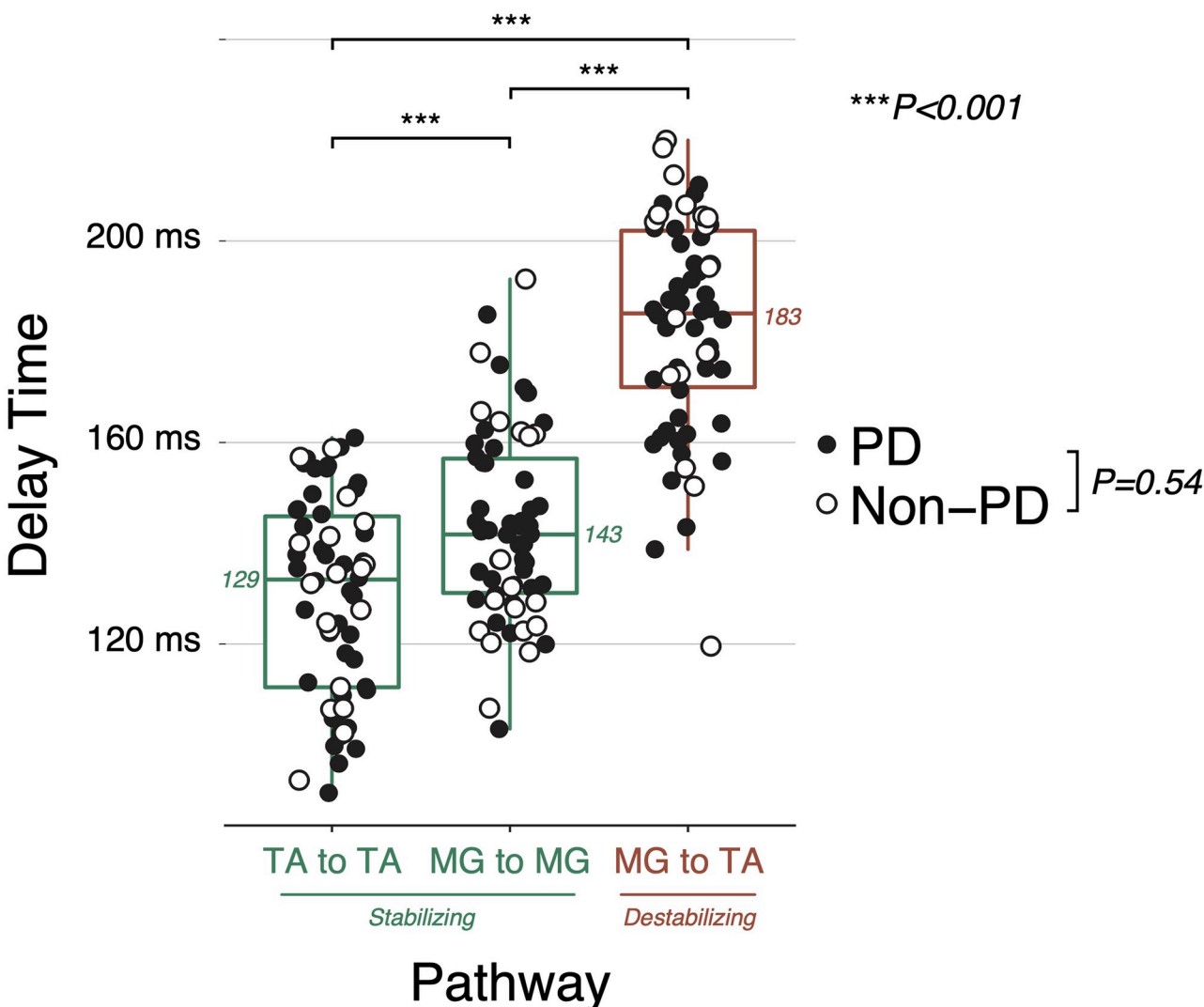

**Fig 5. Comparison of identified SRM delay times between destabilizing and stabilizing feedback pathways.** P values are derived from linear mixed models. N = 62.

muscle activity in response to CoM motion. We show that in older adults with and without PD, the sensorimotor control of agonist muscles exhibits characteristic changes compared to that in young healthy individuals. Further, in older adults with PD, antagonist muscles also exhibit large transient responses when shortened that are time-locked to CoM motion. This abnormal activity presumably hinders balance corrections by stiffening the joints and reducing postural robustness. Although we cannot assess cause-effect relationships in this observational study, the presence and magnitude of abnormal antagonist activity is associated with the number of previous falls, which would be the case if this activity were a cause of falls. Among the clinical variables examined, the strongest association was observed between abnormal antagonist activity and disease duration.

This suggests that CoM sensorimotor control in PD may continue to degenerate after the initial degeneration of dopaminergic neurons in the basal ganglia is largely complete. At least 80% of dopaminergic cells are gone by the time of diagnosis in most PD patients. However, we found an effect of PD duration among patients for whom this process was in later stages, and

for whom there were almost certainly no dopaminergic cells remaining. Therefore, we speculate that these processes continue after dopaminergic cell loss is complete.

**PD does not grossly impact the sensorimotor processes used to activate prime mover muscles during balance control**. Our results show that temporal patterns of agonist muscle activity in older adults with and without PD exhibit a precise relationship to sensory inflow during balance perturbations. Consistent with previous descriptions of EMG activity in PD, [11, 12] we found that active sensorimotor feedback control of agonist muscles (TA and MG) was generally very similar in PD compared to healthy older adults. We used the same sensorimotor transformation previously demonstrated in younger adults and in animals to successfully reproduce the timing and magnitude of muscle activity based on the acceleration, velocity, and displacement of the CoM elicited during perturbations. Contrary to the idea that postural responses in PD are delayed, we found no differences in the delay between CoM motion and muscle activity in older adults with and without PD. In fact, the delays were slightly shorter than those found previously in younger adults. This shorter delay may reflect increased sensory drive due to the increased difficulty of the task for older adults.

**Abnormal antagonist TA activity in PD appears to be generated by temporally precise sensorimotor signals arising from and driving prime mover muscles, but routed differently through the nervous system and with a slightly longer delay**. An important distinction from agonist muscle activation–in which muscles are activated by sensory signals originating in the same muscles [2, 16–18]–is that the antagonist TA response depended on sensorimotor feedback originating in different muscles initially lengthened by the perturbations. To explain this antagonist activity, we added additional feedback channels arising from the agonist MG to describe a polysynaptic feedback arc to the antagonist TA. The sensitivity of the antagonist TA to the acceleration of the CoM revealed the largest differences in muscle activity between older adults with and without PD. Importantly, these signals could not have originated within TA, which is slack during the initial portion of this perturbation. Instead, they likely arise from muscle spindle proprioceptive signals [2, 52] within the MG and within other muscles within the limbs–and potentially torso and other areas–that are stretched due to the perturbations and which exhibit similar feedback-mediated responses.

**Our results suggest that falls in PD may result from abnormal antagonist activity that interferes with otherwise appropriate responses, rather than the inability to activate prime movers**. Postural reactions in PD are often described as "slower" than in individuals without PD, [53] a useful description that is consistent with well-documented delays in reaction time PD patients demonstrate in many tasks [54–56]. However, our results favor the more nuanced description that postural responses are "slow[er] to develop force" than in controls, [53] because we found that the most prominent feature of postural responses in PD was the abnormal activation of antagonist muscles that rendered the ongoing balance correcting response ineffective, rather than absent or delayed responses in agonist muscles. We found that antagonist sensitivity to CoM acceleration feedback was significantly associated with both the presence of fall history and the number of previous falls in the 6 months prior to study enrollment, even after controlling for fall risk factors (age, sex, and FOG). Increased antagonist muscle activity could hinder the generation of corrective joint torques required to stabilize the CoM during standing as well as voluntary movements [57, 58]. The same mechanisms for stabilizing the CoM are likely used during, as corrective muscular responses following perturbations during locomotion are superimposed upon the ongoing locomotor pattern in both the stance and swing limb [59]. Abnormal CoM control may therefore represent a fall risk factor in addition to other factors such as freezing of gait, [33, 60] rigidity, [61] and cognitive impairment [62].

One explanation for these data could be that networks enabling the routing from agonist to antagonist muscles exist in the healthy nervous system, and are selectively

**disinhibited during specific task conditions in youth and healthy aging, but continually disinhibited in PD**. Typically, the control of antagonistic pairs of muscles such as TA and MG is organized reciprocally, so that activation of one inhibits the other [63]. However, in healthy individuals, physiological mechanisms exist that enable co-contraction between these muscles, [64] particularly in unpredictable [17] or unstable [65] environments, and in situations of increased postural threat, such as while standing at an elevated height [66]. Possibly this pathway is disinhibited only when coactivation is necessary in the healthy nervous system, but that is disinhibited more broadly in PD. Top-down "gating" of sensory input has been identified throughout the mammalian nervous system [67]. In PD, whether the sensorimotor control of antagonist muscles is abnormal in general is unknown. However, previous results in other muscles [11, 12] have shown abnormal overall activity similar to that observed here in TA. Notably, although we did not find a robust increase in antagonist activity in MG in coarse analyses of overall muscle activity–and therefore did not perform the entire SRM model fit on this data–anecdotally, some patients exhibited strong antagonist responses in this muscle. Some antagonist activity in MG would be expected for concordance with previous results. Individual cases with and without abnormal antagonist activity in MGAS are described in S1 File.

**Taken together with other studies, these results suggest that sensorimotor control in PD may continue to degenerate due to changes in subcortical mechanisms even after the initial degeneration of dopaminergic neurons in the basal ganglia is largely complete**. There are two aspects of these results that point to a neurophysiological substrate other than the basal ganglia for these deficits. First, these patients were fairly advanced (10 y), and we found a strong effect of PD duration. At this duration, the changes in the basal ganglia are long over. Therefore, to find an effect of duration suggests that there is dependence on pathological mechanisms that continue to progress after 5–10 years. These are largely non-dopaminergic [68, 69]. Second, based on other studies using similar patient cohorts, it is very likely that patients with high values of antagonist activity had considerable cholinergic deficits in regions including the thalamus [70–72]. The latency on the antagonist pathway we identified is consistent with involvement of supraspinal centers.[56] In particular, longer-latency stretch responses in TA ($>95$ ms) can be modulated with transcranial magnetic stimulation, which suggests involvement of supraspinal centers [51]. The involvement of the thalamus in particular is suggested by animal work, which provides some evidence that the thalamus is necessary for the appropriate generation of postural response muscle activity [21] and receives monosynaptic sensory feedback from the spinal cord [73].

## Limitations

There are several limitations to this study of note. First, the lack of imaging data prevents us from more concretely identifying the neuroanatomical substrates of these deficits. Although the ability to image changes in the basal ganglia and other brain regions associated with Parkinson's disease remains limited (cf. [70, 71, 74, 75]), without any such data we are forced to speculate on the neuroanatomical substrates involved. These results suggest that investigations incorporating the balance testing approaches used here in addition to neuroimaging approaches are warranted. Second, the use of retrospective measurements of fall frequency allows the possibility that previous falls could have caused the abnormalities in CoM control we report. Ongoing prospective studies will guard against this possibility. Third, all testing was performed in the OFF medication state, which may limit generalizability of these results to balance challenges and falls that occur during the multiple ON periods of the medication cycle experienced by most people with PD. Although it has been reported that up to 70% of falls

occur while ON medications, [3] other studies have shown that OFF state measurements have more validity to falls [76, 77]. Complicating matters further, medications can improve or impair different gait and balance features, [78] potentially in a patient-specific manner [33]. Testing in both the OFF and ON states could provide more insight into the extent to which the postural abnormalities shown here vary over daily periods of higher and lower fall risk. Additionally, although we showed associations with overall fall frequency, the extent to which deficits in the standing balance testing paradigm used here is valid to predict falls that occur during specific circumstances–such as while standing vs. other tasks such as those including backward perturbations or gait–is unknown. More complex musculoskeletal models may be required to comprehensively evaluate the impact of impaired sensorimotor feedback on fall risk in PD.

## Conclusions

These results demonstrate that the sensorimotor feedback control of agonist muscles is affected by healthy aging, and that the sensorimotor feedback control of antagonist muscles is affected by PD. Abnormal sensorimotor feedback control of antagonist muscles may be a potential cause of falls in PD. Abnormal sensorimotor feedback control of antagonist muscles is associated with increased progression in PD, and may involve non-dopaminergic centers. Clinical evaluations of balance in PD in neurological testing involving involuntary perturbations of the CoM forward with respect to the ankles could reveal important features of impaired balance.

## Materials and methods

### Recruitment

Participants with PD were recruited from healthcare centers and patient advocacy organizations in the Atlanta area. Healthy participants were recruited from older adult advocacy groups, referral from researchers at Emory University and Georgia Tech, and from flyers placed on college campuses. All PD patients met the following inclusion criteria: Hoehn and Yahr Stages I-IV, age $\geq$ 35 years, ability to walk with or without assistive device $\geq$ 10 feet, normal perception of vibration and light touch on feet. Exclusion criteria for PD patients were: significant neurological or musculoskeletal impairment as determined by the authors. Older Non-PD participants were recruited to be similar in age and sex to the PD group, but were not matched individually to each patient. Younger Non-PD participants were recruited from flyers placed on college campuses. Exclusion criteria for all healthy participants were: neurological condition or significant musculoskeletal impairment as determined by the authors. Data were collected from January 2014-July 2018. Some participants (23/44 PD, 11/18 Non-PD) were recruited as part of a rehabilitation study, other outcomes of which have been and will be reported separately [11].

### Ethics statement

All participants provided written informed consent prior to enrollment in accordance with protocols approved by the institutional review boards of Emory University and the Georgia Institute of Technology.

### Assessment protocol

All participants were assessed with common clinical measures of balance ability, PD severity and with a brief cognitive screen [41]. PD patients were instructed to abstain from

antiparkinsonian medications for at least 12 hours prior to assessment but to continue taking all other medications on their typical schedule. PD severity was assessed with the original or MDS-revised Unified Parkinson Disease Rating Scale, part III (MDS UPDRS-III), and items from part II (activities of daily living) relevant to tremor and postural and gait disability [79]. Assessments were videotaped and rated by a movement disorders specialist (author S.A.F.) or MDS certified rater (author M.E.H.). UPDRS-III scores were converted into equivalent MDS UPDRS-III scores according to methods established in the literature [80]. Individual UPDRS-III items were used to establish Tremor-dominant (TD-PD) or Postural Instability and Gait Difficulty (PIGD-PD) PD phenotype [31]. Participants with PD were classified as "freezers" based on scores $\geq 2$ on the Freezing of Gait questionnaire (FOG-Q) [81] item 3, indicating freezing episodes "about once per week. N = 3 patients for whom FOG-Q was unavailable were classified based on UPDRS II item 14, indicating "occasional freezing when walking." Behavioral balance outcomes included the MiniBESTest, [36] Berg Balance Scale (BBS), [40] Fullerton Advanced Balance Scale (FAB), [36] and Dynamic Gait Index (DGI) [39]. Global cognitive status was assessed with the Montreal Cognitive Assessment (MoCA) [35]. Non-PD and PD participants were interviewed with a standardized instrument for health history including the presence of previous falls [11]. Clinical information was abstracted from medical records for 21/44 PD patients; for the remainder of patients and for all neurotypical participants this information was obtained via self-report.

## Medical and other exclusions

Data were initially available for N = 66 participants with PD and N = 32 Non-PD participants. PD participants were excluded from analyses due to: MoCA score < 25 indicative of MCI [35] (N = 10); altered diagnosis after enrollment (N = 5); inability to complete assessment (N = 4); equipment problems leading to invalid EMG or other laboratory data (N = 3). Non-PD participants were excluded from analyses due to: MoCA score < 25 indicative of MCI (N = 5); neurological condition disclosed after enrollment (N = 1); inability to complete assessment (N = 1); equipment problems leading to invalid EMG or other laboratory data (N = 7). After applying exclusions, data of N = 44 PD patients and n = 18 neurotypical participants were available for analysis. Complete case analyses were used in the event of missing data.

## Reactive balance assessments

Reactive balance assessments were conducted with methodology used previously in earlier studies of PD patients [11]. Participants stood barefoot on two force plates installed in a translating platform with their arms crossed across their chest, feet parallel and eyes open and focused on a large landscape poster 4.6 m ahead. Participants were exposed to between 36 and 60 ramp-and-hold translations of the support surface (peak acceleration: 0.1 g; peak velocity: 25 cm/s; peak displacement: 7.5 cm; time from initial acceleration to initial deceleration 450 ms) with direction selected randomly among 12 directions evenly distributed in the horizontal plane and unpredictable by the participant. Stance width was fixed at 26 cm between the medial malleoli. Kinematic, kinetic, and EMG data were collected and synchronized as in previous studies [11, 16–18]. EMG was recorded bilaterally from muscles in the legs and trunk and processed off-line (high-pass, 35 Hz; de-mean; rectify). EMG and other analog signals were sampled at either 1080 Hz or 1200 Hz depending on equipment. Body segment kinematic trajectories were collected at 120 Hz. CoM displacement and velocity in the horizontal plane were calculated from kinematic data as a weighted sum of segmental masses, and CoM acceleration in the horizontal plane was calculated from recorded horizontal-plane forces.

## Data processing

Analyses were conducted on bilateral recordings from *tibialis anterior* (TA) and *medial gastrocnemius* (MG). After high-pass filtering, full-wave rectification, and low-pass filtering, EMG, kinematic, and kinetic signals were aligned to perturbation onset and averaged over replicates of each perturbation direction separately for each participant [11, 16–18]. Additional details are provided in S1 File. The resulting average EMG recordings of each muscle of each participant (TA from the left and right leg as well as MG from the left and right leg) then were normalized to the maximum value observed during a wide window 80–425 ms after perturbation onset in all perturbation directions examined. This window encompassed medium- and long-latency postural reflex responses. Analyses were performed to perturbation directions within ±30˚ of the cardinal anterior and posterior directions, which elicit strong bilateral responses in both muscles studied.

## Sensorimotor response modeling

To quantify whether abnormalities in surface electromyographic activity associated with Parkinson disease reflected central changes in the sensorimotor transformation between center of mass kinematics and recorded muscle activity, we computed relationships between measured patterns of electromyogram magnitude and timing with recorded center of mass kinematic signals using our sensorimotor response model [2, 16–18].

Sensorimotor response model parameters that best reproduced the entire time course of muscle activity were found by minimizing an error term calculated between recorded EMG and reconstructed signals. The error term was quantified as the sum of squared errors at each time sample and the maximum observed error:

$$\min_{k,\lambda}\left\{ \mu_s \int_0^{t_{end}} e^2 dt + \mu_m \max(|e|) + \mu_k k^T k \right\}$$

where first term penalizes squared error $e^2$ between averaged and simulated muscle activity with weight $\mu_s$, the second term penalizes the maximum error between simulated and recorded muscle activity at any point with weight $\mu_m$, and the third term $\mu_k$ is a nuisance term that penalizes the magnitudes of gain parameters $k$ in order to improve convergence when feedback channels do not contribute to reconstructed electromyogram signals. The ratio of weights $\mu_s{:}\mu_m{:}\mu_k$ was 1:1:1e-6. Additional details are provided in S1 File.

**Balance-correcting CoM feedback.** To test whether feedback rules used to active muscles in response to perturbations in the healthy nervous system were altered in PD, we compared the ability of two primary models to reproduce muscle activation patterns based on CoM motion. In both, the overall hypothesis was that CoM kinematic signals are linearly combined in a feedback manner to generate muscle activity.

In the first model, balance-correcting CoM feedback, recorded EMG responses were reconstructed using kinematic signals describing horizontal plane CoM acceleration (*a*), velocity (*v*), and displacement (*d*), that were each weighted by a feedback gain ($k_a$, $k_v$, $k_d$), summed, and subjected to a common time delay ($\lambda$) to simulate neural transmission and processing time:

$$EMG_{recon} = \lfloor k_d d(t - \lambda) + k_v v(t - \lambda) + k_a a(t - \lambda) \rfloor \qquad (1)$$

with the total summed signal subjected to a rectification nonlinearity in order to represent excitatory drive to motor pools:

$$\lfloor \cdot \rfloor = max(0, \cdot)$$

For TA, the signals (*a*, *v*, *d*) describe motion of the CoM backward with respect to the ankles, which cause TA to lengthen, and are hypothesized to be encoded primarily in TA muscle spindles [82]. Conversely, for MG, the signals describe motion of the CoM forward with respect to the ankles, which cause MG to lengthen, and are hypothesized to be encoded primarily in MG muscle spindles. In the model, this is implemented by multiplying kinematic signals recorded in the extrinsic coordinate system of the laboratory by an appropriate factor (1 or -1 given the default coordinate system in our laboratory) so that motion backward with respect to the ankle corresponds to positive values of *a* for TA and so that motion forward with respect to the ankle corresponds to positive values of *a* for MG. We assumed transient acceleration encoding was limited by muscle spindle cross-bridge cycling, [83, 84] and implemented this by setting acceleration feedback to zero after a fixed time window [17, 18] (see S1 File).

**Nonspecific CoM feedback.**  Because the early antagonist TA activity in PD during perturbations when the CoM was initially displaced forward showed some striking similarities to the patterns of magnitude and timing observed in the agonist MG, we also tested a second feedback scenario in which the kinematic signals that are typically used to activate MG could explain TA magnitude and timing. In this scenario, we added additional feedback channels for signals describing motion of the CoM forward with respect to the ankles, which cause TA to shorten (as well as MG to lengthen), and are hypothesized to be encoded primarily in MG muscle spindles:

$$EMG_{recon} = \lfloor k_d d(t - \lambda) + k_v v(t - \lambda) + k_a a(t - \lambda) \rfloor + \lfloor -k_d' d(t - \lambda') - k_v' v(t - \lambda') - k_a' a(t - \lambda') \rfloor \tag{2}$$

In addition to the components of Eq 1, additional delayed feedback gains ($k_a'$, $k_v'$, $k_d'$) and an independent delay $\lambda'$ have been added to describe these additional signals. For TA, signals (*a*, *v*, and *d*) describe motion of the CoM backward with respect to the ankle that lengthen TA, and signals (−*a*, −*v*, and −*d*) describe motion of the CoM forward with respect to the ankle that lengthen MG.

## Statistical methodology

Statistical tests were performed in Matlab r2018b, SAS University Edition 7.2, or R 3.6.1. Tests were considered statistically-significant at P≤0.050. Tests of different kinematic variables (e.g., peak CoM acceleration or peak CoM velocity) or of different model parameters (e.g., $k_a'$ or $\lambda$) were assumed to evaluate independent null hypotheses and were performed without adjustment for simultaneous inference [85]. Summary statistics are presented as sample mean±sample standard deviation, sample mean (95% confidence interval), or count (percent).

**Participants and setting.**  Comparisons of clinical and demographic variables between groups were performed with independent samples *t*-tests and chi-squared tests.

**Differences in peak muscle activity and CoM kinematics across groups.**  Differences in peak muscle activity and CoM kinematics across groups were assessed with ANOVAs with a group factor (HYA vs. Non-PD vs. PD) and with a participant factor included as a random factor nested within group. Each observation entered into ANOVA was the average of all trials of a given perturbation direction for each participant. Significant initial F tests were followed post-hoc subgroup F tests comparing 1) PD vs. Non-PD, and 2) HYA vs. older (PD or Non-PD). P values from post-hoc tests were adjusted using a Holm-Bonferroni sequential procedure [86]. Separate ANOVAs were performed for each variable for forward and backward perturbation directions.

**Differences in SRM parameters across groups.** Differences in SRM parameters across groups were assessed with one-way ANOVAs (HYA vs. Non-PD vs. PD). Each observation corresponded to an individual participant. Significant initial F tests were followed with Holm-Bonferroni-adjusted post-hoc independent-samples *t*-tests comparing: 1) PD vs. Non-PD, and, 2) HYA vs. older (PD or Non-PD). Separate batteries of ANOVA and post-hoc tests were performed for each SRM parameter.

**Fall history classification.** Participants were classified as having 0, 1, or $\geq 2$ falls in the 6 months prior to study enrollment. For older participants, fall history was obtained via self-report on the day of testing. Falls were defined as "an event that results in a person coming to rest unintentionally on the ground or another lower level," a shortened version of an existing definition [6, 87]. Young healthy participants were coded as free of falls for primary analyses. Analyses were iterated with and without young healthy participants included to evaluate sensitivity. One participant for whom fall history data were missing was excluded from analyses involving fall history.

**Associations between fall history and SRM parameters.** Associations between TA SRM parameter $k_a'$ and fall history were assessed in two ways. Multivariate ANOVA assessed variation in SRM parameters with fall classification as described above. Negative binomial regression assessed association between $k_a'$ and the number of falls over the prior 6 months among those participants for whom these data were available. Both approaches included covariates associated with fall risk: increased age, female sex, presence of PD, and presence of FOG [11].

**Associations between SRM parameters and clinical and demographic variables.** We summarized associations between $k_a'$ and clinical variables with Pearson product-moment correlation coefficients. Correlation coefficients were classified as negligible or non-negligible according to criteria proposed by Cohen [49].

**Delay differences between stabilizing and destabilizing pathways.** Identified SRM delay parameters for each participant were entered into a linear mixed model with fixed effects of Pathway (TA-TA, MG-MG, and MG-TA) and PD and a random effect for participant using the lmerTest package in R software.

## Supporting information

**S1 File. Additional analyses.**
(PDF)

## Author Contributions

**Conceptualization:** J. Lucas McKay, Madeleine E. Hackney, Lena H. Ting.

**Data curation:** J. Lucas McKay.

**Formal analysis:** J. Lucas McKay.

**Funding acquisition:** J. Lucas McKay, Lena H. Ting.

**Investigation:** J. Lucas McKay, Kimberly C. Lang, Sistania M. Bong, Madeleine E. Hackney, Stewart A. Factor.

**Methodology:** J. Lucas McKay, Lena H. Ting.

**Project administration:** J. Lucas McKay.

**Visualization:** J. Lucas McKay.

**Writing – original draft:** J. Lucas McKay, Kimberly C. Lang, Sistania M. Bong.

**Writing – review & editing:** J. Lucas McKay, Kimberly C. Lang, Sistania M. Bong, Madeleine E. Hackney, Stewart A. Factor, Lena H. Ting.

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
