## [Decision Letter · Decision Letter 0]

9 Apr 2021

PONE-D-21-04464

Abnormal center of mass control during balance: a new biomarker of falls in people with Parkinson's disease

PLOS ONE

Dear Dr. McKay,

Thank you for submitting your manuscript to PLOS ONE. After careful consideration, we feel that it has merit but does not fully meet PLOS ONE’s publication criteria as it currently stands. Therefore, we invite you to submit a revised version of the manuscript that addresses the points raised during the review process.

Two experts in the field have carefully reviewed the manuscript.

Their comments are appended below.

Both of them acknowledged the manuscript is well written with leaving several concerns which should be clarified before publication.

I would like you to revise the manuscript according to the critiques.

I am sorry to be late for this decision letter.

We look forward to receiving your revised manuscript.

Kind regards,

Manabu Sakakibara, Ph.D.

Academic Editor

PLOS ONE

Journal Requirements:

Reviewers' comments:

Reviewer's Responses to Questions

**Comments to the Author**

1. Is the manuscript technically sound, and do the data support the conclusions?

Reviewer #1: Yes

Reviewer #2: Yes

2. Has the statistical analysis been performed appropriately and rigorously? 

Reviewer #1: Yes

Reviewer #2: Yes

3. Have the authors made all data underlying the findings in their manuscript fully available?

Reviewer #1: Yes

Reviewer #2: Yes

4. Is the manuscript presented in an intelligible fashion and written in standard English?

Reviewer #1: Yes

Reviewer #2: Yes

5. Review Comments to the Author

Reviewer #1: This is a well-written manuscript and had important additions to the text due to the reviewers' suggestion. So I have few questions.

There were no differences in the COM kinematics during the perturbation responses in persons with PD vs. non-PD. As the authors explain the title: “Abnormal center of mass control during balance”? The title suggests that there are different COM kinematics for the PD.

How were the perturbations randomized? How does this affect the results? PD postural responses are known to be context dependent and they are able to modify their strategy trial-by-trial, based on error feedback control.

Reviewer #2: As Reviewer #1 on the prior submission, I thank the authors for their thorough responses to the original review. I reiterate that this is an interesting manuscript that will be of interest to multiple fields interested in balance control and Parkinson's disease. I also thank the authors for their helpful explanations regarding some of their analyses in response to my previous review.

My primary suggestion remains that authors tone down the language regarding a causal link between their modeling results/observations of abnormal TA activity and falls/balance impairment in PD. In my prior review, I did not intend to assert that the authors need to observe falls in the laboratory to demonstrate causality between their neurophys data and balance impairment in PD. One could observe instability in COM kinematics without inducing a fall, yet there was no observable change in COM kinematics in the PD participants who exhibited the aberrant muscle activity.

It is certainly clear that persons with PD show an aberrant muscle response to backward perturbations, and I think this alone is a valuable contribution that is certainly worth further study. However, it is not clear to me from the current data that this aberrant response contributes to instability or falls in daily life. As previously mentioned, the relationships between 6 mo fall frequency and ka' appear to be driven by perhaps one or two participants with very high numbers of falls. I sympathize with the authors regarding the difficulty in using these data in persons with PD given the high heterogeneity, but one must wonder about the robust nature of this relationship given the current data. I also appreciate that the authors have expanded on the limitations of the study, but the findings regarding similarity in muscle responses among groups in the forward perturbations and lack of ON meds data add additional uncertainty.

I want to emphasize again that I enjoyed reading this paper and do not wish to further preclude its publication. I simply find the claims that a new biomarker of falls in persons with PD has been discovered to be a bridge too far given these initial findings in what I expect will eventually become a more extensive line of research.

6. PLOS authors have the option to publish the peer review history of their article (what does this mean?). If published, this will include your full peer review and any attached files.

Reviewer #1: **Yes: **Daniel Boari Coelho

Reviewer #2: No

---

## [Author Response · Author response to Decision Letter 0]

27 Apr 2021

Response to Reviewers

We thank the reviewers for their comments. We believe that we have addressed them. A summary of the changes is described below. 

Reviewer #1

This is a well-written manuscript and had important additions to the text due to the reviewers' suggestion. So I have few questions.

There were no differences in the COM kinematics during the perturbation responses in persons with PD vs. non-PD. As the authors explain the title: “Abnormal center of mass control during balance”? The title suggests that there are different COM kinematics for the PD.

RESPONSE:

To highlight that the feedback responses are abnormal, whereas the CoM kinematics are unchanged, compared to those without PD, we have altered the title from:

"Abnormal center of mass control during balance: a new biomarker of falls in people with Parkinson's disease"

To:

“Abnormal center of mass feedback responses during balance: a potential biomarker of falls in Parkinson’s disease” 

How were the perturbations randomized? How does this affect the results? PD postural responses are known to be context dependent and they are able to modify their strategy trial-by-trial, based on error feedback control.

RESPONSE:

The perturbations were delivered in computer-generated random order, described in the methods as:

"ramp-and-hold translations of the support surface (peak acceleration: 0.1 g; peak velocity: 19

25 cm/s; peak displacement: 7.5 cm; time from initial acceleration to initial deceleration 450 ms) with direction selected randomly among 12 directions evenly distributed in the horizontal plane and unpredictable by the participant."

We acknowledge the context-dependence of postural strategies in PD, and would expect to see additional adaptation/learning effects if they were not delivered in random order. We have not speculated on this in this work as we intend to investigate this in a follow-on study, using a methodology similar to our previous work on trial-by-trial adaptation in young healthy individuals.1

Reviewer #2: As Reviewer #1 on the prior submission, I thank the authors for their thorough responses to the original review. I reiterate that this is an interesting manuscript that will be of interest to multiple fields interested in balance control and Parkinson's disease. I also thank the authors for their helpful explanations regarding some of their analyses in response to my previous review.

My primary suggestion remains that authors tone down the language regarding a causal link between their modeling results/observations of abnormal TA activity and falls/balance impairment in PD. In my prior review, I did not intend to assert that the authors need to observe falls in the laboratory to demonstrate causality between their neurophys data and balance impairment in PD. One could observe instability in COM kinematics without inducing a fall, yet there was no observable change in COM kinematics in the PD participants who exhibited the aberrant muscle activity.

RESPONSE: 

We believe that the majority of this language was edited in the substantial revision following the first round of reviews. In response to this comment, we further reviewed the language for material concerning causality. In addition to modifying the title as described above to use the phrase “a POTENTIAL biomarker of falls,” we have also changed the main body text.

We have altered the statement:

“The presence and magnitude of abnormal antagonist activity is associated with the number of previous falls, which would be the case if this activity were a cause of falls.”

To:

“ALTHOUGH WE CANNOT ASSESS CAUSE-EFFECT RELATIONSHIPS IN THIS OBSERVATIONAL STUDY, the presence and magnitude of abnormal antagonist activity is associated with the number of previous falls, which would be the case if this activity were a cause of falls.”

We have also altered the statement:

“Abnormal sensorimotor feedback control of antagonist muscles is a potential cause of falls in PD.”

To:

“Abnormal sensorimotor feedback control of antagonist muscles MAY BE a potential cause of falls in PD.”

It is certainly clear that persons with PD show an aberrant muscle response to backward perturbations, and I think this alone is a valuable contribution that is certainly worth further study. However, it is not clear to me from the current data that this aberrant response contributes to instability or falls in daily life. As previously mentioned, the relationships between 6 mo fall frequency and ka' appear to be driven by perhaps one or two participants with very high numbers of falls. I sympathize with the authors regarding the difficulty in using these data in persons with PD given the high heterogeneity, but one must wonder about the robust nature of this relationship given the current data.

RESPONSE: 

We thank the reviewer for their time and careful comments. We disagree that a small number of participants drive the relationship between 6 mo fall frequency and ka’ shown in Figure 3C. This relationship results from a negative binomial regression specifically designed to accommodate “overdispersed” data like fall frequency.2–4 However, although we believe that the analysis in Figure 3C is appropriate for these data, we acknowledge that negative binomial regression may be unfamiliar to some readers. Therefore, we also show a relationship between ka’ and fall number discretized as 0, 1, or ≥2 with a more familiar ANOVA in Figure 3B1. Because of this we believe that the association between ka’ and falls shown here is quite robust.

I also appreciate that the authors have expanded on the limitations of the study, but the findings regarding similarity in muscle responses among groups in the forward perturbations and lack of ON meds data add additional uncertainty.

RESPONSE:

We acknowledge this concern and believe that the material previously added to the limitations section adequately discusses these limitations. We are in the process of developing the OFF/ON testing paradigms and expanded perturbation sets that we believe will provide additional insight into these questions. Unfortunately, these studies have been delayed by COVID.

I want to emphasize again that I enjoyed reading this paper and do not wish to further preclude its publication. I simply find the claims that a new biomarker of falls in persons with PD has been discovered to be a bridge too far given these initial findings in what I expect will eventually become a more extensive line of research.

RESPONSE:

We thank the reviewer for this comment. We have altered the language “new biomarker” in the title to “potential biomarker” and made other changes as described above.

References

1. Welch, T. D. & Ting, L. H. Mechanisms of motor adaptation in reactive balance control. PloS one 9, e96440 (2014).

2. Ullah, S., Finch, C. F. & Day, L. Statistical modelling for falls count data. Accident Analysis & Prevention 42, 384–392 (2010).

3. Gill, D. P., Zou, G. Y., Jones, G. R. & Speechley, M. Comparison of regression models for the analysis of fall risk factors in older veterans. Ann Epidemiol 19, 523–530 (2009).

4. Simpson, L. A., Miller, W. C. & Eng, J. J. Effect of Stroke on Fall Rate, Location and Predictors: A Prospective Comparison of Older Adults with and without Stroke. PLoS ONE 6, e19431 (2011).

---

## [Decision Letter · Decision Letter 1]

11 May 2021

Abnormal center of mass feedback responses during balance: a potential biomarker of falls in Parkinson's disease

PONE-D-21-04464R1

Dear Dr. McKay,

We’re pleased to inform you that your manuscript has been judged scientifically suitable for publication and will be formally accepted for publication once it meets all outstanding technical requirements.

Kind regards,

Manabu Sakakibara, Ph.D.

Academic Editor

PLOS ONE

Additional Editor Comments (optional):

Reviewers' comments:

Reviewer's Responses to Questions

**Comments to the Author**

1. If the authors have adequately addressed your comments raised in a previous round of review and you feel that this manuscript is now acceptable for publication, you may indicate that here to bypass the “Comments to the Author” section, enter your conflict of interest statement in the “Confidential to Editor” section, and submit your "Accept" recommendation.

Reviewer #1: All comments have been addressed

Reviewer #2: All comments have been addressed

2. Is the manuscript technically sound, and do the data support the conclusions?

Reviewer #1: Yes

Reviewer #2: Yes

3. Has the statistical analysis been performed appropriately and rigorously? 

Reviewer #1: Yes

Reviewer #2: Yes

4. Have the authors made all data underlying the findings in their manuscript fully available?

Reviewer #1: Yes

Reviewer #2: Yes

5. Is the manuscript presented in an intelligible fashion and written in standard English?

Reviewer #1: Yes

Reviewer #2: Yes

6. Review Comments to the Author

Reviewer #1: (No Response)

Reviewer #2: The authors have revised the manuscript and addressed all comments. I recommend the manuscript for publication.

7. PLOS authors have the option to publish the peer review history of their article (what does this mean?). If published, this will include your full peer review and any attached files.

Reviewer #1: **Yes: **Daniel Boari Coelho

Reviewer #2: No

---

## [Editor Report · Acceptance letter]

17 May 2021

PONE-D-21-04464R1 

Abnormal center of mass feedback responses during balance: a potential biomarker of falls in Parkinson’s disease 

Dear Dr. McKay:

I'm pleased to inform you that your manuscript has been deemed suitable for publication in PLOS ONE. Congratulations! Your manuscript is now with our production department. 

Kind regards, 

on behalf of

Dr. Manabu Sakakibara 

Academic Editor

PLOS ONE